# ComLoRA: A Competitive Learning Approach for Enhancing LoRA

**Qiushi Huang**[1,2]**, Tom Ko**[3]**, Lilian Tang**[2]**, Yu Zhang**[1]*
[1]Southern University of Science and Technology, [2]University of Surrey, [3]ByteDance
{qiushi.huang.cs, tomkocse, yu.zhang.ust}@gmail.com,
h.tang@surrey.ac.uk

## Abstract

We propose a Competitive Low-Rank Adaptation (ComLoRA) framework to address the limitations of the LoRA method, which either lacks capacity with a single rank-$r$ LoRA or risks inefficiency and overfitting with a larger rank-$Kr$ LoRA, where $K$ is an integer larger than 1. The proposed ComLoRA method initializes $K$ distinct LoRA components, each with rank $r$, and allows them to compete during training. This competition drives each LoRA component to outperform the others, improving overall model performance. The best-performing LoRA is selected based on validation metrics, ensuring that the final model outperforms a single rank-$r$ LoRA and matches the effectiveness of a larger rank-$Kr$ LoRA, all while avoiding extra computational overhead during inference. To the best of our knowledge, this is the first work to introduce and explore competitive learning in the context of LoRA optimization. The ComLoRA's code is available at https://github.com/hqsiswiliam/comlora.

## 1 Introduction

Large Language Models (LLMs) have transformed various natural language processing tasks by leveraging their vast number of parameters and advanced architectures (Radford et al., 2019; Achiam et al., 2023; Touvron et al., 2023; Dubey et al., 2024). Despite their success, efficiently adapting LLMs to specific tasks remains a challenge due to the prohibitive costs of full fine-tuning (FFT). Parameter-Efficient Fine-Tuning (PEFT) methods, such as Low-Rank Adaptation (LoRA) (Hu et al., 2021), have emerged as promising solutions by updating only a small portion of parameters, significantly reducing computational burdens.

While LoRA provides an efficient alternative for adapting LLMs, it faces challenges in balancing model expressiveness with parameter efficiency. A single low-rank LoRA module may lack sufficient representational capacity to handle complex tasks, whereas simply increasing the rank introduces additional parameters and the risk of overfitting. Empirical studies on PEFT methods have shown that higher-rank configurations do not consistently outperform lower-rank setups in practice (Hu et al., 2021; Sidahmed et al., 2024; Kalajdzievski, 2023), indicating that merely adding more parameters is not a straightforward solution for performance enhancement. Therefore, there is a need for training strategies that can better exploit existing LoRA components to enhance expressiveness without increasing computational overhead during inference.

Existing LoRA training methods typically employ fixed hyperparameters and adaptation strategies across different tasks and throughout the training process. This lack of adaptability may hinder the model's ability to adjust to tasks with varying levels of complexity, such as those requiring deeper reasoning, context understanding, or handling of specialized terminology. Consequently, the model may fail to strike an optimal balance between efficiency and expressiveness, especially in more demanding tasks (Lialin et al., 2023). This limitation leads to the underperformance of LoRA modules when applied to a diverse range of tasks that challenge the model's capacity to represent complex patterns and relationships in the data.

---

*Corresponding author.

To address these challenges, we draw inspiration from competitive learning, which has been applied successfully in feature discovery and clustering (Nowlan, 1989; Rumelhart & Zipser, 1985; Grossberg, 1987). Competitive learning allows multiple components of a model to compete during training, dynamically optimizing their performance based on feedback. Despite its successes in other domains, this mechanism has not yet been explored within the context of LoRA or other PEFT methods.

We propose the **Competitive Low-Rank Adaptation (ComLoRA)** framework, which incorporates competitive learning to dynamically train multiple LoRA components. In ComLoRA, multiple LoRA components compete during adaptation, guided by a dynamic selector that evaluates their performance at each training step. This competitive process allows each component to improve iteratively, achieving an optimal balance between expressiveness and parameter efficiency without significantly increasing computational overhead during inference. By leveraging the strengths of the most effective LoRA components, ComLoRA enhances model adaptation across diverse tasks while maintaining inference efficiency.

Our key contributions are as follows:

- We propose the first competitive learning framework for LoRA.

- We develop a selector mechanism that optimizes LoRA component selection during training.

- Extensive experiments demonstrate the superiority of ComLoRA over LoRA, without increasing inference overhead.

## 2 RELATED WORKS

**LoRA.** Low-Rank Adaptation (LoRA) (Hu et al., 2021) is a widely used technique for efficient fine-tuning of large pre-trained models by introducing low-rank updates to parameter matrices, thereby reducing computational overhead. While LoRA maintains strong performance with fewer parameters, its fixed rank can limit expressiveness. Attempts to increase the rank, such as using $K \times r$, enhance capacity but at the cost of higher computational demands and increased risk of overfitting. ComLoRA addresses these challenges by incorporating a competitive mechanism where multiple LoRA components compete during training, enhancing adaptability and robustness without sacrificing the efficiency that defines LoRA.

**Mixture of Experts (MoE).** MoE models have gained significant attention in the context of large language models (LLMs) due to their ability to scale efficiently by selectively activating only a subset of the model's parameters during inference. A prominent example is the Switch Transformer (Fedus et al., 2022), which uses an MoE layer to route inputs to different expert networks, significantly reducing computational costs while maintaining performance. Similarly, GLaM (Du et al., 2022) dynamically selects a subset of experts based on the input, allowing the model to adapt efficiently to diverse tasks. MoELoRA (Luo et al., 2024) extends the MoE framework to LoRA by using a gating network to route inputs to multiple LoRA experts during inference, emphasizing diversity among components. In contrast, ComLoRA focuses on learning a single LoRA component after competitive training, minimizing inference overhead. Hence, MoELoRA needs to utilize multiple LoRA experts and the gating network during inference, while the proposed ComLoRA method acts exactly the same as LoRA during the inference.

## 3 METHODOLOGY

In this section, we introduce the Competitive Low-Rank Adaptation (ComLoRA) framework.

### 3.1 OVERVIEW

ComLoRA initializes $K$ distinct LoRA components, each with a rank $r$. A **LoRA selector** is introduced to dynamically choose the most suitable LoRA component based on the input context. This competitive framework drives the LoRA components to refine their individual strengths, effectively

Figure 1: An illustration of the ComLoRA pipeline.

combining them to enhance the model's overall performance without adding inference overhead. An illustration of the proposed ComLoRA method is shown in Figure 1.

## 3.2 MULTIPLE LORA COMPONENTS

We begin by initializing $K$ separate LoRA components $\{\text{LoRA}_1, \text{LoRA}_2, \ldots, \text{LoRA}_K\}$, each with rank $r$. For $\text{LoRA}_k$, we have projection matrices $\mathbf{A}_k \in \mathbb{R}^{d \times r}$ and $\mathbf{B}_k \in \mathbb{R}^{r \times l}$, where $d$ and $l$ denotes the dimension of the input and output in this layer, respectively. The matrices $\mathbf{A}_k$ are initialized using the Kaiming initialization method (He et al., 2015), while the matrices $\mathbf{B}_k$ are initialized to zero, following standard LoRA practices.

## 3.3 LORA SELECTOR

The **LoRA selector** is designed to evaluate the input sequence and select the most suitable LoRA component. It operates by computing similarity scores between a representation of the input sequence and embeddings associated with each LoRA component.

**Input Representation.** Given an input sequence $\mathbf{X} \in \mathbb{R}^{L \times d}$ of length $L$, where each token is embedded in a $d$-dimensional space, we obtain a contextualized representation using a lightweight one-layer transformer encoder. This neural network produces hidden states $\mathbf{H} \in \mathbb{R}^{L \times d_{\text{sel}}}$, where $d_{\text{sel}}$ is the dimensionality of the selector's hidden states. To obtain a fixed-length representation of the input sequence, we average the hidden states across the sequence length as

$$\mathbf{h}_{\text{avg}} = \frac{1}{L} \sum_{i=1}^{L} \mathbf{H}_i, \tag{1}$$

where $\mathbf{H}_i$ is the hidden state corresponding to the $i$-th token.

**LoRA Embeddings.** Each LoRA component $k$ is associated with a learnable embedding vector $\mathbf{e}_k \in \mathbb{R}^{d_{\text{sel}}}$. These embeddings serve as representatives for their respective LoRA, allowing the selector to compute and update the similarity scores efficiently.

**Computation of Similarity Score.** The similarity score between the input representation and each LoRA component embedding is calculated using a dot product:

$$\text{sim}_k = \mathbf{h}_{\text{avg}}^{\top} \mathbf{e}_k \ \text{ for } k = 1, 2, \ldots, K. \tag{2}$$

This results in a similarity score vector $\mathbf{s} \in \mathbb{R}^K$ as

$$\mathbf{s} = [\text{sim}_1, \text{sim}_2, \ldots, \text{sim}_K]. \tag{3}$$

## 3.4 OBJECTIVE FUNCTION

The objective function to train ComLoRA consists of three losses, including the language modeling loss, selector loss, and alignment loss. In the following section, we introduce them one by one.

**Language Modeling Loss.** The language modeling loss, denoted as $\mathcal{L}_{\text{LM}}$, is calculated to guide the updates of the LoRA components. For each LoRA component $k$, the loss is defined as:

$$\mathcal{L}_{\text{LM},k} = -\sum_{t=1}^{L} \log P(y_t | y_{<t}, \theta, \mathbf{A}_k, \mathbf{B}_k), \ \mathcal{L}_{\text{LM}} = \frac{1}{K} \sum_{k=1}^{K} \mathcal{L}_{\text{LM},k}, \tag{4}$$

where $y_t$ is the target token at position $t$, $\theta$ represents the base model parameters, and $\mathbf{A}_k$ and $\mathbf{B}_k$ are the projection matrices of the $k$-th LoRA component. This loss function drives the training process, ensuring that the LoRA components are optimized to minimize the prediction error of the next token in the sequence.

**Alignment Loss.** To ensure that the LoRA selector effectively influences the language model (LM) during training, we introduce an alignment loss that reflects the selector's preferences in the LM updates. This alignment loss is designed to align the similarity scores, as predicted by the selector, with the next-token prediction losses for the top-$N$ ranked LoRA components. The loss is defined as:

$$\mathcal{L}_{\text{align}} = \eta(t) \cdot \left( \text{StopGrad} \left( \mathbf{p}_{\text{sim, top-}N} \right)^\top \mathcal{L}_{\text{LM, top-}N} \right), \tag{5}$$

where $\mathbf{p}_{\text{sim, top-}N}$ denotes the normalized similarity scores of the top-$N$ selected LoRA components with largest similarity scores, $\mathcal{L}_{\text{LM, top-}N}$ is a vector containing LM losses for the top-$N$ selected LoRA components, $\text{StopGrad}(\cdot)$ indicates that the input is detached to prevent it from contributing to the gradient, and $\eta(t)$ is an annealing scalar that evolves over training steps. By focusing on the top-N components, the alignment loss directs LM updates based on the top preferences of the LoRA selector. With the annealing strategy, we expect that during the initial stage of the training process, LoRA components can learn more independently to encourage exploration and as training progresses, the influence of the guidance from the LoRA selector on the LoRA components gradually increases.

**Pairwise Loss.** The LoRA selector is trained to align the similarity scores with the actual performance of the LoRA components using a pairwise loss approach. This ensures that the similarity scores correlate with the relative performance (as measured by the language modeling loss) of the LoRA components. To avoid overly deterministic selection, a Gaussian noise term is added to the language modeling loss for each LoRA component. The modified LM loss for component $k$ is defined as:

$$\tilde{\mathcal{L}}_{\text{LM},k} = \mathcal{L}_{\text{LM},k} + \alpha \epsilon_k, \tag{6}$$

where $\epsilon_k \sim \mathcal{N}(0,1)$ is a Gaussian noise term, and $\alpha$ is a small constant that controls the noise intensity (e.g., $\alpha = 0.1$). The differences between similarity scores and the corresponding noisy LM losses for each pair of LoRA components $(i, j)$ are defined as:

$$\Delta_{\text{sim},i,j} = \text{sim}_i - \text{sim}_j, \ \Delta_{\text{LM},i,j} = \tilde{\mathcal{L}}_{\text{LM},i} - \tilde{\mathcal{L}}_{\text{LM},j}. \tag{7}$$

Those differences are normalized using the softmax operation to produce log-softmax values as

$$\text{logsoftmax}(\Delta_{\text{sim}}) = \log(\text{softmax}(\Delta_{\text{sim}})), \ \text{logsoftmax}(\Delta_{\text{LM}}) = \log(\text{softmax}(\Delta_{\text{LM}})). \tag{8}$$

The pairwise loss is then computed as the $\ell_1$ loss between the log-softmax similarity score differences and the log-softmax LM loss differences:

$$\mathcal{L}_{\text{pairwise}} = \|\text{logsoftmax}(\Delta_{\text{sim}}) - \text{StopGrad}(\text{logsoftmax}(\Delta_{\text{LM}}))\|_1, \tag{9}$$

where $\| \cdot \|_1$ denotes the $\ell_1$ norm of a vector. This pairwise loss encourages the selector to assign higher similarity scores to LoRA components that demonstrate better performance (i.e., lower noisy LM losses), ensuring that the predictions of the selector align with the actual capabilities of each component.

**Objective Function.** The total loss combines the LM loss, the alignment loss, and the selector loss as

$$\mathcal{L}_{\text{total}} = \mathcal{L}_{\text{LM}} + \mathcal{L}_{\text{align}} + \mathcal{L}_{\text{pairwise}}. \tag{10}$$

During training, we optimize $\mathcal{L}_{\text{total}}$ with respect to the parameters in LoRA components and the LoRA selector.

## 3.5 DETERMINATION OF LoRA WINNER

After training, we use the LoRA selector to determine the LoRA winner based on the validation dataset. Specifically, we pass the validation data through the LoRA Selector to compute the similarity scores for each LoRA component as

$$\mathbf{s}_{\text{val}} = [\text{sim}_{\text{val},1}, \text{sim}_{\text{val},2}, \ldots, \text{sim}_{\text{val},K}], \tag{11}$$

where $\text{sim}_{\text{val},k}$ is the accumulated similarity score for LoRA component $k$ over the validation dataset.

We then select the LoRA winner as the one with the highest total similarity score as

$$k^{\text{win}} = \arg \max_{k} \text{sim}_{\text{val},k}. \tag{12}$$

During inference on the test set, we only use the LoRA winner $\text{LoRA}_{k^{\text{win}}}$, without using the LoRA selector or other LoRA components. This ensures that the inference is efficient, as it avoids any additional computational overhead associated with the selector or dynamic selection.

Table 1: Comparison of Full Fine-Tuning (FFT), LoRA, MoE of LoRA, and ComLoRA.

|                          | FFT    | LoRA | MoE of LoRA | ComLoRA |
|--------------------------|--------|------|-------------|---------|
| Overfitting Risk         | High   | Low  | Medium      | Low     |
| High Inference Overhead  | Low    | Low  | High        | Low     |
| Competitive Learning     | Low    | Low  | Low         | High    |
| Model Complexity         | High   | Low  | High        | Medium  |

### 3.6 COMPARISONS WITH FFT, LORA, AND MOE OF LORA

During the training process, we can see that the proposed ComLoRA method needs to train $K$ LoRAs, leading to training complexity comparable to MoE with $K$ LoRAs but higher than LoRA. During the inference process, the proposed ComLoRA method only uses the LoRA winner, which can be merged into the base model. Hence, ComLoRA preserves low inference overhead as LoRA did, while MoE of LoRA, which needs to use a gating network to choose from $K$ LoRAs, cannot be merged into the base model, yielding additionally computational costs. In summary, the comparisons with FFT, LoRA, and MoE of LoRA are shown in Table 1.

## 4 EXPERIMENTS

In this section, we evaluate the proposed ComLoRA method by comparing with LoRA to compare between the competitive training in ComLoRA and the conventional training process in LoRA.

### 4.1 EXPERIMENTAL SETTINGS

All experiments were conducted using the LLaMA-3-8B model (Dubey et al., 2024). To compare ComLoRA with LoRA across different rank configurations, we use LoRA ranks of 4, 8, 16, 32, and 128, and ComLoRA configurations with $(K = 4, r = 4)$, $(K = 2, r = 8)$, $(K = 4, r = 8)$, and $(K = 4, r = 32)$.

The AdamW optimizer (Loshchilov & Hutter, 2019) is used to train LoRA and ComLoRA. The learning rates for both LoRA and ComLoRA methods are selected from $[1e - 3, 1e - 4]$, and fine-tuning is conducted for 3 epochs for all the tasks. Each experiment runs with 5 different seeds, and the average results are reported.

### 4.2 EVALUATION TASKS

**Commonsense Reasoning.** We evaluate our models using a comprehensive commonsense reasoning dataset, which includes eight sub-tasks: BoolQ (Clark et al., 2019), PIQA (Bisk et al., 2020), SIQA (Sap et al., 2019), HellaSwag (Zellers et al., 2019), WinoGrande (Sakaguchi et al., 2021), ARC-c (Clark et al., 2018), ARC-e (Clark et al., 2018), and OBQA (Mihaylov et al., 2018). These tasks cover various aspects of commonsense knowledge, such as physical reasoning, social implications, and natural language inference. We aggregate the training sets from all sub-tasks into a single corpus of 170,420 entries, from which we randomly select 120 entries for validation to identify the optimal model.

**MMLU.** The MMLU (Massive Multitask Language Understanding) benchmark (Hendrycks et al., 2021b;a) includes 57 diverse subjects spanning the humanities, STEM, social sciences, and more. Each subject comprises questions of varying difficulty, with multiple-choice answers provided.

Table 3: Performance of LoRA and ComLoRA on commonsense reasoning tasks. The Params (%) (L/S) column indicates the percentage of trainable parameters in LoRA component(s) (denoted by L) and the LoRA selector (denoted by S). The best performance is in **bold**, and the second best performance is underlined.

| Method | Params (%) (L/S) | BoolQ | PIQA | SIQA | ARC-c | ARC-e | OBQA | HellaS | WinoG | Average |
|---|---|---|---|---|---|---|---|---|---|---|
| LoRA$_{r=4}$ | 0.09/0.00 | 65.72 | 73.67 | 73.49 | 63.31 | 78.70 | 72.80 | 80.06 | 75.93 | 72.96 |
| LoRA$_{r=8}$ | 0.18/0.00 | 52.02 | 70.40 | 64.94 | 59.04 | 76.39 | 64.20 | 74.72 | 72.38 | 66.76 |
| LoRA$_{r=16}$ | 0.35/0.00 | 65.66 | 77.64 | 71.65 | 71.93 | 87.42 | 73.60 | 82.06 | 73.40 | 75.42 |
| LoRA$_{r=32}$ | 0.70/0.00 | 70.80 | 85.20 | 79.90 | 71.20 | 84.20 | 79.00 | 91.70 | 84.30 | 80.79 |
| LoRA$_{r=128}$ | 2.74/0.00 | 71.99 | 88.19 | 79.63 | 80.03 | 91.37 | 86.40 | 93.42 | 88.00 | 84.88 |
| ComLoRA$_{K=4,r=4}$ | 0.35/1.45 | 69.02 | 86.56 | 79.68 | 78.84 | 92.21 | 83.40 | 92.39 | 86.35 | 83.56 |
| ComLoRA$_{K=2,r=8}$ | 0.35/1.45 | **75.14** | **90.04** | **82.09** | 82.42 | **93.14** | **89.00** | **96.03** | **88.87** | **87.09** |
| ComLoRA$_{K=4,r=8}$ | 0.70/1.45 | 73.79 | 88.79 | 81.47 | **83.11** | 92.68 | **89.00** | 95.45 | 88.24 | 86.57 |
| ComLoRA$_{K=4,r=32}$ | 2.74/1.39 | 71.53 | 87.65 | 79.99 | 80.12 | 92.42 | 87.40 | 94.56 | 86.50 | 85.02 |

**Personalized Conversation Task.** We assess model performance on personalized conversational understanding using the CONVAI2 dataset (Dinan et al., 2019; Zhang et al., 2018). This dataset is designed to evaluate a model's ability to engage in meaningful and coherent conversations while maintaining a personalized dialogue. Following (Liu et al., 2020; Song et al., 2021; Huang et al., 2023b;a; 2024), we use a self-persona configuration that reveals only the speaker's persona. The task involves training the model to respond in a way that reflects an understanding of personal preferences and contextual nuances.

## 4.3 EVALUATION METRICS

To evaluate performance on the commonsense reasoning datasets, we use accuracy as the primary metric, following the approach of (Hu et al., 2023). For each test instance, the language models generate answers based on the provided queries, and specific keywords (e.g., "true" or "false" for BoolQ) are searched within the responses. The first occurrence of a relevant keyword is recorded as the model's answer, while responses lacking relevant keywords are considered incorrect. A similar evaluation method is employed for MMLU.

For the CONVAI2 dataset, we assess linguistic similarity using BLEU (Papineni et al., 2002), METEOR (Banerjee & Lavie, 2005), and ROUGE-L (R-L) (Lin, 2004), which calculate the overlap of n-grams between model predictions and ground truth. To measure semantic similarity, we use BERT Score (Zhang et al., 2019), which evaluates the cosine similarity of normalized BERT embeddings between predictions and ground truth. We report BERT Score's F1 ($BERT_{F1}$), Recall ($BERT_R$), and Precision ($BERT_P$).

Table 2: Performance comparison of LoRA and ComLoRA on MMLU. The best performance is in **bold**, and the second best performance is underlined.

| Method | Params (L/S) (%) | Accuracy |
|---|---|---|
| LoRA$_{r=4}$ | 0.09/0.00 | 56.44 |
| LoRA$_{r=8}$ | 0.18/0.00 | 56.79 |
| LoRA$_{r=16}$ | 0.35/0.00 | 55.25 |
| LoRA$_{r=32}$ | 0.70/0.00 | 55.97 |
| LoRA$_{r=128}$ | 2.74/0.00 | 59.36 |
| ComLoRA$_{K=2,r=8}$ | 0.35/1.45 | 59.81 |
| ComLoRA$_{K=4,r=4}$ | 0.35/1.45 | 59.41 |
| ComLoRA$_{K=4,r=8}$ | 0.70/1.45 | 59.14 |
| ComLoRA$_{K=4,r=32}$ | 2.74/1.39 | **61.09** |

## 4.4 RESULTS

**Commonsense Reasoning.** The results on commonsense reasoning are shown in Table 3. Among the LoRA configurations, LoRA$_{r=128}$ achieves the highest average score of 84.88%, indicating that a higher rank enhances the model capacity to learn complex patterns in the data. In contrast, the ComLoRA framework shows strong performance across all tested configurations. Notably, ComLoRA$_{K=2,r=8}$, which selects the winner LoRA with rank 8 for inference, achieves the highest average score of 87.09%, outperforming all LoRA baselines, including LoRA$_{r=128}$. This result underscores the effectiveness of ComLoRA's competitive learning mechanism, where multiple low-rank adapters compete during training, leading to improved overall performance.

Moreover, ComLoRA configurations with lower ranks still outperform higher-rank LoRA. For example, ComLoRA$_{K=4,r=4}$ achieves an average score of 83.56%, surpassing LoRA$_{r=32}$ which has an average score of 80.79%. This demonstrates that ComLoRA can achieve superior performance with fewer parameters compared to conventional LoRA methods.

Table 4: Performance comparison of LoRA and ComLoRA on CONVAI2 Dataset. The best performance is in **bold**, and the second best performance is underlined.

| Method | Params (L/S) (%) | BLEU | METEOR | R-L | BERT$_{F1}$ | BERT$_R$ | BERT$_P$ | Average |
|---|---|---|---|---|---|---|---|---|
| LoRA$_{r=4}$ | 0.09/0.00 | 2.54 | 12.97 | 12.27 | 84.76 | 85.04 | 84.54 | 47.02 |
| LoRA$_{r=8}$ | 0.18/0.00 | 2.37 | 12.69 | 11.89 | 84.66 | 84.97 | 84.40 | 46.83 |
| LoRA$_{r=16}$ | 0.35/0.00 | 3.15 | 14.00 | 13.32 | 84.70 | 84.25 | 85.19 | 47.43 |
| LoRA$_{r=32}$ | 0.70/0.00 | 3.27 | 15.06 | 14.05 | 84.78 | 84.41 | 85.19 | 47.79 |
| LoRA$_{r=128}$ | 2.74/0.00 | 3.24 | 15.24 | 14.06 | 84.79 | 84.42 | 85.20 | 47.82 |
| ComLoRA$_{K=2,r=8}$ | 0.35/1.45 | **3.69** | **17.05** | **16.09** | **85.23** | **84.74** | **85.75** | **48.76** |
| ComLoRA$_{K=4,r=4}$ | 0.35/1.45 | 3.67 | 16.99 | 15.98 | 85.22 | 84.73 | 85.74 | 48.72 |
| ComLoRA$_{K=4,r=8}$ | 0.70/1.45 | 3.65 | 16.90 | 15.95 | 85.19 | 84.70 | 85.72 | 48.69 |
| ComLoRA$_{K=4,r=32}$ | 2.74/1.39 | 3.62 | 16.06 | 14.75 | 84.83 | 84.53 | 85.17 | 48.16 |

For individual tasks, ComLoRA$_{K=2,r=8}$ consistently outperforms different LoRA methods. In tasks such as PIQA and HellaSwag, ComLoRA$_{K=2,r=8}$ achieves remarkable scores of 90.04% and 96.03%, respectively, significantly higher than the corresponding performance of LoRA$_{r=128}$. This indicates that the competition among LoRA components in ComLoRA enable the model to learn from complex reasoning tasks.

**MMLU.** As shown in Table 2, ComLoRA demonstrates superior performance across various configurations. Notably, ComLoRA$_{K=4,r=32}$ achieves the highest accuracy of 61.09%, surpassing all LoRA baselines, including LoRA$_{r=128}$, which achieves 59.36%. ComLoRA$_{K=2,r=8}$ secures the second-best accuracy at 59.81%, followed closely by ComLoRA$_{K=4,r=4}$ with 59.41%. These results highlight the effectiveness of ComLoRA's competitive learning approach, as it consistently outperforms traditional LoRA methods even with lower ranks per component. Additionally, ComLoRA$_{K=4,r=8}$ maintains strong performance, further demonstrating ComLoRA's robustness across various configurations.

**CONVAI2.** According to the results shown in Table 4, ComLoRA$_{K=2,r=8}$ outperforms all other configurations across multiple metrics, including BLEU, METEOR, ROUGE-L (denoted by R-L), and BERT scores (i.e., BERT$_{F1}$, BERT$_R$, and BERT$_P$). Specifically, ComLoRA$_{K=2,r=8}$ achieves a BLEU score of 3.69, a METEOR score of 17.05, and a ROUGE-L score of 16.09, significantly higher than the corresponding scores from LoRA$_{r=4}$ to LoRA$_{r=128}$. This demonstrates the effectiveness of competitive learning approach in ComLoRA to capturing diverse conversational patterns.

Additionally, ComLoRA$_{K=4,r=4}$ secures the second-best performance across all metrics, maintaining high scores and further validating the effectiveness of the ComLoRA framework. The average scores reflect ComLoRA's ability to consistently outperform traditional LoRA, even with lower ranks, thereby offering a more parameter-efficient alternative without sacrificing model performance.

## 5 ABLATION STUDIES

To thoroughly evaluate the effectiveness of the proposed ComLoRA framework, we conduct a series of ablation studies on commonsense reasoning tasks. Unless otherwise specified, all experiments are performed using the LLaMA-3-8B model with $K = 4$ and $r = 8$ in ComLoRA.

### 5.1 ANALYSIS ON LOSS AND LORA COMPONENTS

To assess the effectiveness of ComLoRA, we analyze its training losses when $K = 4$ and $r = 4$ and compare with LoRA$_{r=4}$ and LoRA$_{r=16}$. As shown in Figure 2, the training losses of all LoRA components in ComLoRA are consistently lower than those of LoRAs, despite each LoRA component having only a rank of 4. Different LoRA components in ComLoRA intertwine, collectively forming lower losses through their competitive interaction during training. This results in that ComLoRA achieves a faster convergence compared to a single LoRA model under the standard fine-tuning process.

Figure 3 plots the number of times being the winner among the four LoRA components for each LoRA component over every 50 steps, where the winner at a training step has the lowest training

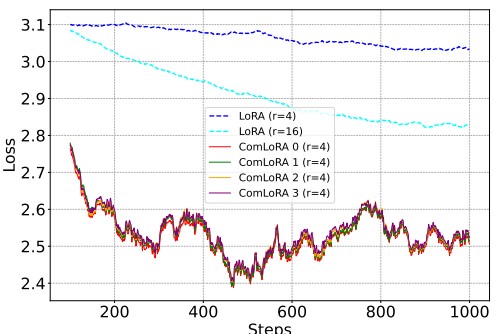

Figure 2: Loss Curves for LoRA and Com-LoRA.

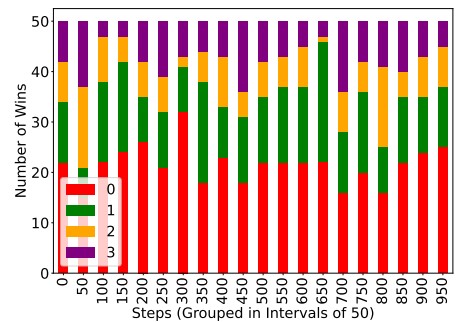

Figure 3: Number of times being the winner for each LoRA component inside ComLoRA every 50 training steps.

Table 5: Impact of removing one of the losses or selector from total loss.

| Method | BoolQ | PIQA | SIQA | ARC-c | ARC-e | OBQA | HellaS | WinoG | Average |
|---|---|---|---|---|---|---|---|---|---|
| ComLoRA$_{K=4,r=4}$ | 69.02 | 86.56 | 79.68 | 78.84 | 92.21 | 83.40 | 92.39 | 86.35 | 83.56 |
| w/o $\mathcal{L}_{\text{LM}}$ | 67.31 | 84.82 | 76.97 | 77.99 | 91.08 | 79.80 | 91.23 | 80.82 | 81.25 |
| w/o $\mathcal{L}_{\text{align}}$ | 67.46 | 79.16 | 75.28 | 69.80 | 81.65 | 78.20 | 83.07 | 80.90 | 76.94 |
| w/o $\mathcal{L}_{\text{pairwise}}$ | 69.20 | 85.69 | 74.21 | 77.56 | 90.87 | 78.60 | 89.86 | 77.51 | 80.44 |
| w/o Selector | 65.38 | 83.84 | 70.98 | 74.74 | 90.07 | 75.20 | 72.24 | 71.03 | 75.44 |

loss. Figure 3 reveals how the LoRA component 0 gradually gains dominance through competitive training, ultimately being the LoRA winner in ComLoRA.

## 5.2 EFFECT OF DIFFERENT LOSSES AND SELECTOR

To better understand the impact of each loss component and the selector in the ComLoRA framework, we conduct ablation studies by removing one component at a time and observing its effect on performance across commonsense reasoning tasks.

According to the results presented in Table 5, we can see that removing the language modeling loss $\mathcal{L}_{\text{LM}}$ leads to a noticeable performance drop, with the average accuracy decreasing from 83.56% to 81.25%. This indicates that the LM loss is essential for optimizing the model's predictions and driving ComLoRA's performance.

The alignment loss $\mathcal{L}_{\text{align}}$, which ensures that the LoRA selector's preferences influence the LM updates, has an even larger impact. Without this loss, the model's average performance decreases significantly to 76.94%, showing the critical role of aligning the selector's guidance with the LM performance.

Similarly, removing the pairwise loss $\mathcal{L}_{\text{pairwise}}$, which fosters competition among LoRA components, also reduces performance, with the average accuracy falling to 80.44%. This suggests that competition between components is important for enhancing the model's performance.

Lastly, we also evaluate the impact of removing the selector entirely from the framework. Without the selector, the average accuracy drops drastically to 75.44%, indicating that the selector plays a central role in ComLoRA's performance.

## 5.3 EFFECT OF THE NUMBER OF LORA COMPONENTS $K$

In this section, we investigate the impact of the number of LoRA components $K$ to the performance of ComLoRA, while fixing the rank $r$ to be 8. According to the results shown in Table 6, we observe that when $K = 1$, which corresponds to the standard LoRA with rank $r = 8$, the average performance across tasks is 66.76%. As we increase $K$, the performance improves significantly, reaching the highest average performance (i.e., **87.09%**) when $K$ equals 2. This demonstrates that introducing competition among multiple LoRA components can substantially enhance the model capability even with a small number of components.

Table 6: Performance of ComLoRA with varying numbers of LoRA components $K$ (with fixed rank $r = 8$) across commonsense reasoning tasks.

| r | K | BoolQ | PIQA | SIQA | ARC-c | ARC-e | OBQA | HellaS | WinoG | Average |
|---|---|-------|------|------|-------|-------|------|--------|-------|---------|
|   | 1 | 52.02 | 70.40 | 64.94 | 59.04 | 76.39 | 64.20 | 74.72 | 72.38 | 66.76 |
|   | 2 | **75.14** | **90.04** | **82.09** | **82.42** | **93.14** | **89.00** | **96.03** | **88.87** | **87.09** |
|   | 3 | 73.06 | 86.29 | 80.40 | 81.57 | 93.10 | 87.60 | 94.75 | 87.61 | 85.55 |
| 8 | 4 | 71.65 | 84.33 | 79.79 | 79.01 | 88.80 | 86.20 | 92.55 | 85.71 | 83.51 |
|   | 5 | 70.64 | 85.80 | 80.30 | 81.91 | 92.34 | 85.80 | 94.12 | 85.79 | 84.59 |
|   | 6 | 68.56 | 84.87 | 79.27 | 77.39 | 91.46 | 80.00 | 93.19 | 83.82 | 82.32 |
|   | 7 | 67.13 | 84.87 | 78.10 | 79.35 | 90.07 | 82.60 | 92.17 | 84.21 | 82.31 |
|   | 8 | 70.86 | 87.11 | 81.06 | 80.38 | 92.09 | 83.60 | 93.46 | 87.45 | 84.50 |

Table 7: Impact of top-$N$ selection to task performance when varying $N$.

| r | K | Top-N | BoolQ | PIQA | SIQA | ARC-c | ARC-e | OBQA | HellaS | WinoG | Average |
|---|---|-------|-------|------|------|-------|-------|------|--------|-------|---------|
|   |   | 1 | 74.19 | 88.47 | 80.86 | 82.76 | 92.13 | 86.80 | 94.81 | 88.87 | 86.11 |
| 8 | 4 | 2 | 72.14 | 88.19 | **82.65** | 82.42 | 92.93 | 88.20 | 95.27 | 87.37 | 86.15 |
|   |   | 3 | 73.49 | **89.39** | 80.81 | 83.28 | 92.85 | 87.80 | 95.38 | 87.06 | 86.26 |
|   |   | 4 | **74.53** | 89.01 | 82.55 | **83.53** | **93.06** | **88.80** | **95.86** | **89.34** | **87.08** |

Interestingly, the performance does not continue to improve with larger values of $K$ beyond 2. While $K = 3$ and $K = 5$ still achieve good average performance (i.e., 85.55% and 84.59%), they are slightly lower than the peak performance at $K = 2$. This suggests that having many competing LoRA components may introduce redundancy or interference to affect the training of each LoRA component.

## 5.4 EFFECT OF $N$ IN TOP-$N$ SELECTION

We evaluate the impact of $N$ in selecting the top-$N$ LoRA components when defining the alignment loss in Eq. (5). According to the results shown in Table 7, we can see that increasing $N$ generally improves the performance, with $N = 4$ achieving the highest average accuracy of 87.08%. This indicates that allowing more LoRA components to update during training enhances the model performance. Moreover, training with top-2 or top-3 components also yields competitive results, providing a balance between the computational efficiency and performance.

## 5.5 EFFECT OF ANNEALING STRATEGY

In this section, we evaluate the impact of the annealing strategy used in the alignment loss (i.e., Eq. (5)) to the performance of ComLoRA. We compare four annealing strategies, including the constant, cosine, exponential, and linear annealing strategy. The four strategies are defined as

$$\text{Constant: } \eta(t) = 1, \quad \text{Cosine: } \eta(t) = 0.5 \cdot \left(1 - \cos\left(\frac{\pi \cdot t}{T}\right)\right) \tag{13}$$

$$\text{Exponential: } \eta(t) = 1 - \exp\left(-\alpha \cdot \frac{t}{T}\right), \quad \text{Linear: } \eta(t) = \frac{t}{T}, \tag{14}$$

where $t$ denotes the index of the current training step and $T$ denotes the number of total steps in the whole training process. According to the results presented in Table 8, we can see that the exponential annealing strategy yields the highest average performance, achieving an average accuracy of 86.57%, which indicates its effectiveness in dynamically adjusting the guidance of the LoRA

Table 8: Performance of different annealing strategies.

| Annealing Strategy | BoolQ | PIQA | SIQA | ARC-c | ARC-e | OBQA | HellaS | WinoG | Average |
|--------------------|-------|------|------|-------|-------|------|--------|-------|---------|
| Constant | 70.21 | 87.43 | 79.84 | 80.12 | 91.58 | 84.00 | 92.83 | 85.79 | 83.98 |
| Cosine | 70.52 | **88.79** | 80.14 | 79.95 | 91.33 | 83.20 | 93.60 | 84.29 | 83.98 |
| Exponential | 73.79 | **88.79** | **81.47** | **83.11** | **92.68** | **89.00** | **95.45** | **88.24** | **86.57** |
| Linear | **74.31** | 87.60 | 80.60 | 81.06 | 91.71 | 86.60 | 94.50 | 86.58 | 85.37 |

Table 9: Impact of varying noise intensity parameter $\alpha$ in the pairwise loss on task performance

| r | K | Noise Intensity ($\alpha$) | BoolQ | PIQA | SIQA | ARC-c | ARC-e | OBQA | HellaS | WinoG | Average |
|---|---|---|---|---|---|---|---|---|---|---|---|
| 8 | 4 | 0.0 | 72.81 | 87.65 | 79.99 | 81.31 | 92.47 | 84.60 | 91.97 | 87.29 | 84.76 |
| | | 0.1 | **73.79** | **88.79** | **81.47** | **83.11** | 92.68 | **89.00** | **95.45** | **88.24** | **86.57** |
| | | 1.0 | 73.09 | 88.14 | 80.19 | 80.80 | 92.21 | 85.20 | 94.55 | 87.21 | 85.18 |
| | | 10.0 | 71.59 | 87.87 | 80.76 | 80.55 | 93.06 | 84.00 | 94.46 | 85.87 | 84.77 |

Table 10: Performance comparison of LoRA components (indexed from 0 to 3) in ComLoRA. The LoRA winner determined by ComLoRA for inference is highlighted as 0.

| LoRA Index | BoolQ | PIQA | SIQA | ARC-c | ARC-e | OBQA | HellaS | WinoG | Average |
|---|---|---|---|---|---|---|---|---|---|
| 0 (Winner) | 73.79 | **88.79** | 81.47 | **83.11** | 92.68 | **89.00** | **95.45** | **88.24** | **86.57** |
| 1 | **74.92** | 88.52 | **82.14** | 81.40 | 92.42 | **89.00** | 95.44 | 87.21 | 86.38 |
| 2 | 73.61 | 88.47 | 81.63 | 82.08 | **92.97** | 87.40 | **95.45** | 87.85 | 86.18 |
| 3 | 74.40 | 88.41 | 81.22 | 81.14 | 91.67 | 86.00 | 95.32 | 86.98 | 85.64 |

selector. The other three annealing strategies exhibit inferior average performance, making the the exponential annealing strategy a good and default choice in our experiments.

## 5.6 EFFECT OF NOISE INTENSITY

To evaluate the impact of noise intensity in the pairwise loss, we conduct an ablation study on the effect of the value of the noise intensity parameter $\alpha$ to the performance. According to the results shown in Table 9, we can see that introducing a moderate level of noise (i.e., $\alpha = 0.1$) yields the best overall performance across tasks, achieving an average accuracy of 86.57%. In contrast, removing the noise entirely (i.e., $\alpha = 0.0$) results in a lower average accuracy of 84.76%, highlighting the importance of controlled noise in improving the generalization. As $\alpha$ increases, the performance generally decreases, indicating that excessive noise (e.g., $\alpha = 10.0$) hinders the model capacity.

## 5.7 EFFECTIVENESS OF THE LORA WINNER

In this section, we examine whether the LoRA winner selected by the ComLoRA framework during training is indeed the best-performing one. Table 10 presents the performance of each LoRA component (indexed from 0 to 3) on commonsense reasoning tasks.

As shown in Table 10, the LoRA winner, indexed as 0, achieves the highest average accuracy of 86.57% across tasks. This confirms that the ComLoRA framework effectively selects the best-performing LoRA component during training. Those results underscore that ComLoRA's selection mechanism reliably identifies the most effective LoRA component, ensuring that the final model benefits from the best available adaptation to the task at hand.

## 6 CONCLUSIONS

In this work, we introduced the ComLoRA framework, a novel approach to enhance LoRA by integrating competitive learning. ComLoRA enhances the efficiency and adaptability of LoRA by initializing multiple LoRA components that compete during training, with each component striving to achieve better performance through competition. The proposed ComLoRA method significantly outperforms the conventional LoRA method across various tasks, while maintaining parameter efficiency and avoiding computational overhead during inference. Our future work will explore the competitive learning paradigm to other fine-tuning methods.

## ACKNOWLEDGEMENTS

This work is supported by National Key R&D Program of China 2022ZD0160300 and NSFC key grant under grant no. 62136005.

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
