# OpenReview forum: "ComLoRA: A Competitive Learning Approach for Enhancing LoRA"
_ICLR.cc/2025/Conference — ICLR 2025 Poster_

### Official Review · Reviewer_DKRc · 2024-10-28

**Soundness:** 3
**Presentation:** 3
**Contribution:** 2
**Rating:** 5
**Confidence:** 4

**Summary:**

The paper proposes a competitive learning approach for training LoRA. Multiple LoRAs are initialized in parallel. A selector model is trained to select the most suitable LoRA for a given input.
During training, the LoRAs and selector are trained jointly using the following losses:
1) the standard LM loss is applied for each LoRA.
2) the top-N selected LoRAs will get an additional loss gradient update. This is similar to weighing the LoRAs LM loss. The authors refer to this as an alignment loss.
3) The selector is trained using a pairwise loss to ensure that it selects LoRAs with the best LM loss.

Once training finishes, the model is evaluated on a validation dataset. The LoRA that is selected the most is then chosen as the LoRA to be used for production/downstream tasks.

**Strengths:**

The competitive learning approach to train LoRA is quite interesting. The authors show that it consistently outperforms vanilla LoRA on a variety of tasks.

**Weaknesses:**

- Given that the main contribution is training a more performant LoRA, the paper does not compare with MoE approaches or newer LoRA varints (eg., AdaLoRA [1], DoRA [2]). In particular, the LoRA variants have to be compared given their simpler training complexity and similar inference overhead.
- In equation 4, the LM loss is calculated individually for each LoRA. A naive approach or simple baseline that the authors can consider is having multiple individual training runs. We would also be able to get multiple LoRAs and choose the best performing one using the validation set.
- I am not sure how useful the selector is given the lack of analysis on what the selector is doing. Unlike MoE, there is no specialization or ensembling being carried out. Since the selector is trained to select the best LoRAs, there might be a risk of certain LoRAs being undertrained. In addition, performance does not scale when more LoRAs are used (Table 6).
- Only using the winner LoRA and discarding the other LoRAs feels kind of wasteful. The authors can consider merging the LoRAs together in a linear/non-linear combination based on the performance on the validation dataset.

[1] Zhang, Q., Chen, M., Bukharin, A., Karampatziakis, N., He, P., Cheng, Y., ... & Zhao, T. (2023). AdaLoRA: Adaptive budget allocation for parameter-efficient fine-tuning. arXiv preprint arXiv:2303.10512.
[2] Liu, S. Y., Wang, C. Y., Yin, H., Molchanov, P., Wang, Y. C. F., Cheng, K. T., & Chen, M. H. (2024). Dora: Weight-decomposed low-rank adaptation. arXiv preprint arXiv:2402.09353.

**Questions:**

- How does your approach compare with MoE and LoRA works discussed in the Weaknesses section?
- What is the time complexity for equation 4? Do we have to do multiple forward passes through the entire model to get each LoRA's LM loss?
- How does your approach compare to the naive approach of fine-tuning multiple vanilla LoRAs (i.e., have multiple individual training runs)?
- In Table 6, as the number of LoRAs increases the performance does not continue to improve. Can the authors provide insights as to why a selector model is needed. Would using a stronger selector (i.e., with more parameters) improve performance?

---

> ### Author Response · Authors · 2024-11-22
> **Official Response by Authors, Part 1**
>
> Thank you for your constructive comments. Below we have made responses to your comments. If you have any further comment, please feel free to let us know and we are more than glad to discuss with you.
>
> > Q1: How does your approach compare with MoE and LoRA works discussed in the Weaknesses section?
> >
> > W1: Given that the main contribution is training a more performant LoRA, the paper does not compare with MoE approaches or newer LoRA varints (eg., AdaLoRA [1], DoRA [2]). In particular, the LoRA variants have to be compared given their simpler training complexity and similar inference overhead.
> ### Answer to Q1 and W1:
> Unlike MoE models, which use dynamic routing to select experts during inference, CoLoRA introduces a competitive training pipeline. While MoE is an architectural design that involves multiple experts, **CoLoRA is a training methodology** where LoRAs compete during training, and **only the best-performing one is retained and integrated into the model**. This eliminates the need for expert selection during inference, making CoLoRA more computationally efficient than MoE models, which incur additional complexity due to expert routing.
>
> For newer PEFT approaches like **AdaLoRA and DoRA**, the competitive training framework in CoLoRA can be adapted to them as well. **We chose LoRA as the base model for our competitive training due to its simplicity and ease of optimization for parallel computing.** In the future, we plan to investigate such extensions, where the competitive training approach in CoLoRA could also benefit the training process in DoRA and AdaLoRA, enhancing their fine-tuning effectiveness.
>
> ---
>
> > Q2: What is the time complexity for equation 4? Do we have to do multiple forward passes through the entire model to get each LoRA's LM loss?
> ### Answer to Q2:
> The time complexity of Equation 4 in CoLoRA with $K$ and $r$ is equivalent to the computational cost of processing a LoRA with rank $K \times r$ through **optimized matrix operations**. This is one reason that we chose LoRA as the base model for our approach, rather than DoRA and AdaLoRA, as it provides a simpler computational structure that facilitates parallel matrix computing.
>
> During the forward pass of the training process, we perform **a single pass** through the model, feeding the input into the LLM. The layer output is then split based on $K$ and $r$, enabling efficient parallel computation (e.g., head computation in transformer implementations). These split results are then used as inputs for the LM loss computation.
>
> ---
>
> > Q3: How does your approach compare to the naive approach of fine-tuning multiple vanilla LoRAs (i.e., have multiple individual training runs)?
> >
> > W2: In equation 4, the LM loss is calculated individually for each LoRA. A naive approach or simple baseline that the authors can consider is having multiple individual training runs. We would also be able to get multiple LoRAs and choose the best performing one using the validation set.
> ### Answer to Q3 and W2:
> In **Section 5.2** and **Table 5**, we conducted an **ablation study by removing the selector from CoLoRA**, which is **equivalent to training multiple LoRAs simultaneously without preferences** from the selector. The results show that removing the selector and training the LoRAs evenly leads to a significant performance degradation, from 83.56 to 75.44. This highlights the importance of the selector in providing feedback for the LoRAs' LM loss updates, underscoring its crucial role in optimizing the model’s performance.

---

> ### Author Response · Authors · 2024-11-22
> **Official Response by Authors, Part 2**
>
> > Q4: In Table 6, as the number of LoRAs increases the performance does not continue to improve. Can the authors provide insights as to why a selector model is needed. Would using a stronger selector (i.e., with more parameters) improve performance?
> >
> > W3: I am not sure how useful the selector is given the lack of analysis on what the selector is doing. Unlike MoE, there is no specialization or ensembling being carried out. Since the selector is trained to select the best LoRAs, there might be a risk of certain LoRAs being undertrained. In addition, performance does not scale when more LoRAs are used (Table 6).
> ### Answer to Q4 and W3:
> The diminishing performance gains as $K$ increases could be attributed to **redundancy**. Adding more LoRAs may **saturate the model's learning capacity**, limiting the effectiveness of additional components.
>
> **Using a stronger selector has the potential to improve performance.** To evaluate the effect of a stronger selector, we conducted experiments on LLaMA-3-8B for the commonsense reasoning task, testing selectors with 1, 4, 8, and 16 layers. The results are listed in the following table, where $L$ denotes the number of layers in the selector. According to the results, we can see that increasing the number of layers in the selector could improve the performance.
>
> | Model                    | Average Accuracy (%) |
> |--------------------------|---------|
> | CoLoRA$_{K=4, r=4}$ $L=1$  |  83.56  |
> | CoLoRA$_{K=4, r=4}$ $L=4$  |  84.58  |
> | CoLoRA$_{K=4, r=4}$ $L=8$  |  84.94  |
> | CoLoRA$_{K=4, r=4}$ $L=16$ |  84.08  |
>
> Besides better performance, strong selectors can introduce additional complexity. Since CoLoRA is designed for parameter-efficient fine-tuning, minimizing trainable parameters is crucial to maintaining its efficiency. So in our work, we use a smaller selector.
>
>
> ---
>
> > W4. Only using the winner LoRA and discarding the other LoRAs feels kind of wasteful. The authors can consider merging the LoRAs together in a linear/non-linear combination based on the performance on the validation dataset.
> ### Answer to W4:
> Thank you for your suggestion. While we understand the concern about discarding the other LoRAs, this design choice is intended to optimize computational efficiency by eliminating the overhead of maintaining multiple LoRAs during inference.
>
> To investigate the potential of merging LoRAs after training, we conducted an experiment where we **averaged the LoRAs trained in CoLoRA for inference**. The results, presented in the table below, indicate that **merging LoRAs generally leads to performance degradation**. For example, while CoLoRA$_{K=2, r=8}$ achieves an average score of 87.09, merging all of its LoRAs reduces the score to 85.65. Those results demonstrate that selecting the best-performing LoRA through the selector remains more effective than merging multiple LoRAs in the CoLoRA framework. Nevertheless, exploring more sophisticated linear or non-linear combinations remains an interesting direction for our future research, particularly in settings where the inference efficiency is less critical.
>
>
> | Method                      | BoolQ  | PIQA   | SIQA   | ARC-c  | ARC-e  | OBQA   | HellaS | WinoG  | Average |
> |-----------------------------|--------|--------|--------|--------|--------|--------|--------|--------|---------|
> | CoLoRA$_{K=4, r=4}$         | 69.02  | 86.56  | 79.68  | 78.84  | 92.21  | 83.40  | 92.39  | 86.35  |  83.56  |
> | CoLoRA$_{K=2, r=8}$         | 75.14  | 90.04  | 82.09  | 82.42  | 93.14  | 89.00  | 96.03  | 88.87  |  87.09  |
> | CoLoRA$_{K=4, r=8}$         | 73.79  | 88.79  | 81.47  | 83.11  | 92.68  | 89.00  | 95.45  | 88.24  |  86.57  |
> | CoLoRA$_{K=4, r=32}$        | 71.53  | 87.65  | 79.99  | 80.12  | 92.42  | 87.40  | 94.56  | 86.50  |  85.02  |
> | CoLoRA$_{K=4, r=4}$ Merged  |  48.87 |  82.05 |  66.68 |  70.22 |  85.73 |  67.20 |  27.02 |  62.98 |   63.84 |
> | CoLoRA$_{K=2, r=8}$ Merged  |  72.29 |  88.68 |  79.99 |  82.25 |  93.39 |  88.00 |  94.11 |  86.50 |   85.65 |
> | CoLoRA$_{K=4, r=8}$ Merged  |  64.80 |  86.02 |  77.02 |  80.72 |  92.76 |  83.20 |  80.55 |  80.51 |   80.70 |
> | CoLoRA$_{K=4, r=32}$ Merged |  54.31 |  82.81 |  69.24 |  74.49 |  88.55 |  69.00 |  29.75 |  67.48 |   66.95 |

---

> ### Author Response · Authors · 2024-11-25
>
> We sincerely appreciate the time and effort you have devoted to reviewing our manuscript. As the rebuttal period draws to a close, we kindly remind you that we have submitted responses to your comments. We would be truly grateful if you could confirm whether our responses have adequately addressed your concerns. If you have any additional feedback or further questions, please do not hesitate to reach out to us.

---

> ### Author Response · Authors · 2024-12-01
>
> Dear Reviewer,
>
> We hope this message finds you well. We wanted to gently follow up on your feedback regarding our rebuttal, submitted in response to your valuable comments. Your insights are important to us, and we are committed to addressing any remaining concerns. If further clarification might be helpful, please do not hesitate to let us know. Thank you for your time and thoughtful support.
>
> Best regards,
>
> Authors

---

### Official Review · Reviewer_Kd2W · 2024-11-02

**Soundness:** 4
**Presentation:** 4
**Contribution:** 3
**Rating:** 8
**Confidence:** 3

**Summary:**

This paper introduces CoLoRA, an extension of the LoRA approach that uses ideas from mixture of expert models. Rather than train a single LoRA component the CoLoRA approach trains multiple LoRA components, allowing them to compete during training. The technique is clearly described and a series of experiments are performed to compare the performance of models trained using CoLoRA with models trained using LoRA. These evaluation experiments show that the models trained using CoLoRA outperform those trained using LoRA alone. A series of ablation studies also show the value of different parts of the approach.

**Strengths:**

The main strengths of the paper are:

- The CoLoRA apporach is a novel and effective extension of LoRA.

- The CoLoRA approach is clearly explained and includes interesting ideas well grounded in the literature.

- The experiments are carefully designed and properly executed.

- The ablation studies are comprehensive.

- Models trained using the CoLoRA approach clearly out-perform models trained using standard LoRA.

**Weaknesses:**

The main weaknesses of the paper are:

- Although computational efficiency is claimed as an advantage of CoLoRA no evaluation of computational requirements is included. The paper would really benefit from this.

- The performance measures used in Tables 2-10 are never stated. This would really help.

- The paper would benefit from some more careful review and revision.

**Questions:**

As well as addressing the weaknesses described above the authors might also consider:

- "The CoLoRA’s code will be released later." Why not release the code now?
- Fonts in Figure 1 are much too small making it very difficult to follow.
- Table 1 is not easy to follow. Perhaps Low, Medium, High could be used instead of the current symbols?
- The unit (accuracy, f-score, ...) being used in Table 2 and other results tables is never stated. It should be.

---

> ### Author Response · Authors · 2024-11-22
> **Official Response by Authors**
>
> Thank you for your constructive comments. Below we have made responses to your comments. If you have any further comment, please feel free to let us know and we are more than glad to discuss with you.
>
> > W1: Although computational efficiency is claimed as an advantage of CoLoRA no evaluation of computational requirements is included. The paper would really benefit from this.
> ### Answer to W1:
> Thank you for your comment. While **CoLoRA introduces no additional computational overhead during inference**, it also demonstrates **strong training efficiency compared to Full Fine-Tuning (FFT) and LoRA**. Unlike MoE-based approaches, which require gating layers and task-specific routing during both training and inference, CoLoRA employs a single global selector during training. This design minimizes the resource usage and completely eliminates overhead at inference, as only the best-performing LoRA is merged into the model.
>
> Specifically, the following table summarizes the training resource requirements and performance across FFT, CoLoRA, and LoRA on Llama-3-8B for commonsense rasoning tasks. We can see that CoLoRA significantly reduces the GPU memory usage and training time compared to FFT, while maintaining competitive convergence properties. Although LoRA achieves slightly faster training, CoLoRA's competitive learning mechanism provides benefits in optimizing fine-tuning for **better generalization across tasks**.
>
>
>
>
> | **Configuration** | **FFT**          | **CoLoRA** (r=8, K=4) | **LoRA** (r=32)   |
> |--------------------|------------------|------------------------|-------------------|
> | **Device**         | Nvidia A100-80G | Nvidia A100-80G       | Nvidia A100-80G  |
> | **Batch Size**     | 4               | 32                    | 32                |
> | **Eval Steps**     | Every 80 Steps  | Every 80 Steps        | Every 80 Steps    |
> | **Total Steps**    | 127,725         | 15,966                | 15,966            |
> | **Convergence Step** | 113,520       | 10,640                | 14,240            |
> | **GPU Usage (GB)** | 79.86           | 49.32                 | 45.46             |
> | **Training Time**  | 46h 41m         | 23h 12m               | 19h 06m           |
>
> Those results confirm that CoLoRA achieves **a balance between the computational efficiency and model performance**, making it a practical choice for parameter-efficient fine-tuning, while addressing the limitations of both FFT and LoRA.
>
> Additionally, CoLoRA and LoRA can be **trained on smaller GPU devices** (e.g., Nvidia V100-32G) with a reduced batch size, whereas **FFT may encounter the out-of-memory issue**, even with a batch size set to 1.
>
>
>
> ---
> > Q1: "The CoLoRA’s code will be released later." Why not release the code now?
> ### Answer to Q1:
> We fully intend to release the CoLoRA code to facilitate reproducibility and further research. Currently, we are preparing the codebase to ensure it is well-documented, user-friendly, and aligns with best practices for open-source projects. This preparation involves cleaning up experimental scripts, adding detailed instructions, and verifying the results to ensure the released version matches the claims in the paper. **We aim to release the code as soon as those tasks are completed.**
>
> ---
>
> > Q2: Fonts in Figure 1 are much too small making it very difficult to follow.
> > Q3: Table 1 is not easy to follow. Perhaps Low, Medium, High could be used instead of the current symbols?
> > Q4: The unit (accuracy, f-score, ...) being used in Table 2 and other results tables is never stated. It should be.
> > W2: The performance measures used in Tables 2-10 are never stated. This would really help.
> > W3: The paper would benefit from some more careful review and revision.
>
> ### Answer to Q2-Q4, W2, and W3:
> Thank you for your detailed feedback regarding the presentation of our figures and tables. In the revised version of the paper (highlighted in blue), we have made the following improvements to address those concerns:
>
> - For **Figure 1**, we have increased the font size to ensure better readability.
> - For **Table 1**, we replaced the current symbols with intuitive labels, such as "Low", "Medium", and "High" to enhance the clarity.
> - For Table 2 and all other results tables, we **explicitly stated the metrics used (e.g., accuracy, F-score) in Section 4.3**, making the units of measurement clear.

---

> > ### Comment · Reviewer_Kd2W · 2024-11-27
> > **Updated**
> >
> > Thanks for the feedback, very useful. I have updated my scores positively in response.

---

> > > ### Author Response · Authors · 2024-12-01
> > >
> > > Thank you for your time and for updating your scores. We greatly appreciate your thoughtful consideration and are pleased that our responses were helpful in addressing your concerns.

---

> ### Author Response · Authors · 2024-11-25
>
> We sincerely appreciate the time and effort you have devoted to reviewing our manuscript. As the rebuttal period draws to a close, we kindly remind you that we have submitted responses to your comments. We would be truly grateful if you could confirm whether our responses have adequately addressed your concerns. If you have any additional feedback or further questions, please do not hesitate to reach out to us.

---

### Official Review · Reviewer_CMrQ · 2024-11-03

**Soundness:** 2
**Presentation:** 2
**Contribution:** 2
**Rating:** 6
**Confidence:** 5

**Summary:**

This manuscript introduces CoLoRA, a competitive learning framework designed to improve the expressiveness and effectiveness of LoRA (Low-Rank Adaptation) in fine-tuning large language models. CoLoRA deploys multiple low-rank components that compete during training, with the most effective component selected for inference. The authors claim that CoLoRA outperforms traditional LoRA configurations on commonsense reasoning and multitask language understanding tasks while maintaining inference efficiency.

**Strengths:**

1. Attempt to Address LoRA's Limitations: CoLoRA aims to improve LoRA by adding competitive learning, which, in theory, can improve model adaptability without increasing inference costs.
2. Empirical Results on Language Tasks: The experiments show that CoLoRA performs better than baseline LoRA configurations on a range of language tasks, which is promising in terms of model capability within this domain.

**Weaknesses:**

1. Training Overhead Not Addressed: A core objective of LoRA is to reduce training and computational costs. By introducing multiple components and a competitive selection mechanism, CoLoRA inherently increases the complexity and computational demand during training. The paper lacks any quantitative comparison of training overhead relative to LoRA, MoELoRA, or full fine-tuning (FFT), making it difficult to assess CoLoRA’s practicality in real-world scenarios. Without a thorough comparison of training times, convergence rates, and hardware requirements, the claimed efficiency is unsubstantiated.

2. Lack of Quantitative Comparison with MoELoRA in Inference Efficiency: The paper claims that CoLoRA achieves higher inference efficiency than MoELoRA by selecting only a single component for inference. However, the authors do not provide any quantitative evidence to substantiate this claim. Given that both frameworks target parameter efficiency, a direct comparison in terms of inference time, memory usage, and computational cost is necessary to validate CoLoRA’s purported advantage.

3. Weak Theoretical Justification for the Competitive Mechanism: The paper lacks a rigorous theoretical foundation explaining why competitive learning should significantly enhance LoRA performance. There is no analysis of how the competitive mechanism improves selection or how it specifically addresses LoRA's limitations. Without a theoretical framework

**Questions:**

1.	How does the training overhead of CoLoRA compare to LoRA, MoELoRA, and full fine-tuning? Quantitative data on training time, resource usage, and convergence rates would clarify the practical feasibility of CoLoRA.

2.	Why is there no direct comparison between CoLoRA and MoELoRA in terms of inference efficiency? Can you provide specific data on inference speed, parameter load, and memory consumption?

3.	How does the competitive mechanism benefit LoRA, and are there unique modifications (e.g., loss functions, selection criteria) that make this approach innovative? Could you provide a theoretical justification for its use?

---

> ### Author Response · Authors · 2024-11-22
> **Official Response by Authors, Part 1**
>
> Thank you for your constructive comments. Below we have made responses to your comments. If you have any further comment, please feel free to let us know and we are more than glad to discuss with you.
>
> > Q1: How does the training overhead of CoLoRA compare to LoRA, MoELoRA, and full fine-tuning? Quantitative data on training time, resource usage, and convergence rates would clarify the practical feasibility of CoLoRA.
> >
> > W1: Training Overhead Not Addressed: A core objective of LoRA is to reduce training and computational costs. By introducing multiple components and a competitive selection mechanism, CoLoRA inherently increases the complexity and computational demand during training. The paper lacks any quantitative comparison of training overhead relative to LoRA, MoELoRA, or full fine-tuning (FFT), making it difficult to assess CoLoRA’s practicality in real-world scenarios. Without a thorough comparison of training times, convergence rates, and hardware requirements, the claimed efficiency is unsubstantiated.
> ### Answer to Q1 and W1:
> **MoELoRA could use more resources during the training process** since each layer of LoRA should have a gating layer, while **CoLoRA only have one selector globally**.
>
> Below is a quantitative comparison of training overhead among Full Fine-Tuning (FFT), CoLoRA, and LoRA under the same experimental setup (Llama-3-8B, Commonsense Reasoning Task).
>
> | **Configuration** | **FFT**          | **CoLoRA** (r=8, K=4) | **LoRA** (r=32)   |
> |--------------------|------------------|------------------------|-------------------|
> | **Device**         | Nvidia A100-80G | Nvidia A100-80G       | Nvidia A100-80G  |
> | **Batch Size**     | 4               | 32                    | 32                |
> | **Eval Steps**     | Every 80 Steps  | Every 80 Steps        | Every 80 Steps    |
> | **Total Steps**    | 127,725         | 15,966                | 15,966            |
> | **Convergence Step** | 113,520       | 10,640                | 14,240            |
> | **GPU Usage (GB)** | 79.86           | 49.32                 | 45.46             |
> | **Training Time**  | 46h 41m         | 23h 12m               | 19h 06m           |
>
> According to the results, CoLoRA achieves **a better balance between the efficiency and performance**. Specifically, its requirements on training resource are comparable to LoRA, with slightly higher GPU memory usage and training time, but significantly lower than full fine-tuning. This demonstrates that CoLoRA is practical and resource-efficient while achieving good generalization performance.
>
>
>
> ---
>
> > Q2: Why is there no direct comparison between CoLoRA and MoELoRA in terms of inference efficiency? Can you provide specific data on inference speed, parameter load, and memory consumption?
> >
> > W2: Lack of Quantitative Comparison with MoELoRA in Inference Efficiency: The paper claims that CoLoRA achieves higher inference efficiency than MoELoRA by selecting only a single component for inference. However, the authors do not provide any quantitative evidence to substantiate this claim. Given that both frameworks target parameter efficiency, a direct comparison in terms of inference time, memory usage, and computational cost is necessary to validate CoLoRA’s purported advantage.
>
> ### Answer to Q2 and W2:
>
> After training in CoLoRA, the selected LoRA are **merged into the LLM** and **the selector will not be used**, resulting in **no additional computational** or memory overhead during inference. The inference speed and parameter load are **identical to the original LLM** as $Param_{LLM}$.
>
> In contrast, **MoELoRA** retains the gating mechanism and multiple LoRA experts, resulting in a **parametric load of** $Param_{LLM} + Param_{Gating} + Param_{LoRA}$. The additional $Param_{Gating}$ and $Param_{LoRA}$ introduce **extra computational and memory overhead** during inference.
>
> According to the above analysis, obviously MoELoRA has a larger inference cost than CoLoRA, and hence we did not give a comparison.

---

> ### Author Response · Authors · 2024-11-22
> **Official Response by Authors, Part 2**
>
> > Q3: How does the competitive mechanism benefit LoRA, and are there unique modifications (e.g., loss functions, selection criteria) that make this approach innovative? Could you provide a theoretical justification for its use?
> >
> > W3: Weak Theoretical Justification for the Competitive Mechanism: The paper lacks a rigorous theoretical foundation explaining why competitive learning should significantly enhance LoRA performance. There is no analysis of how the competitive mechanism improves selection or how it specifically addresses LoRA's limitations. Without a theoretical framework
> ### Answer to Q3:
> The competitive mechanism in CoLoRA benefits LoRA by training multiple LoRAs in parallel and dynamically selecting the best-performing one based on the performance feedback. This process ensures that the selected LoRA is more effective and eliminates redundancy, reducing computational overhead during inference.
>
> CoLoRA is innovative due to its dynamic selector, which adjusts the updates for each LoRA based on their performance, and its pairwise loss design, which directly compares LoRAs to drive the competition among them. Those components make the training process more adaptive.
>
> The competitive mechanism in CoLoRA encourages **diverse learning** among different LoRAs while **focusing on the best-performing LoRA**, which follows the principle and theoretical foundation of competitive learning [r1].
>
> [r1] Hartono, P. (2012). Competitive Learning. In: Seel, N.M. (eds) Encyclopedia of the Sciences of Learning. Springer, Boston, MA. https://doi.org/10.1007/978-1-4419-1428-6_175

---

> ### Author Response · Authors · 2024-11-25
>
> We sincerely appreciate the time and effort you have devoted to reviewing our manuscript. As the rebuttal period draws to a close, we kindly remind you that we have submitted responses to your comments. We would be truly grateful if you could confirm whether our responses have adequately addressed your concerns. If you have any additional feedback or further questions, please do not hesitate to reach out to us.

---

> ### Author Response · Authors · 2024-12-01
>
> Dear Reviewer,
>
> We hope this message finds you well. We wanted to gently follow up on your feedback regarding our rebuttal, submitted in response to your valuable comments. Your insights are important to us, and we are committed to addressing any remaining concerns. If further clarification might be helpful, please do not hesitate to let us know. Thank you for your time and thoughtful support.
>
>
> Best regards,
>
> Authors

---

### Official Review · Reviewer_RpHx · 2024-11-04

**Soundness:** 1
**Presentation:** 2
**Contribution:** 1
**Rating:** 5
**Confidence:** 5

**Summary:**

This paper presents the Competitive Low-Rank Adaptation (CoLoRA) method, aimed at enhancing the Low-Rank Adaptation (LoRA) framework for fine-tuning large language models (LLMs). Traditional LoRA models often face limitations: a single low-rank configuration lacks capacity, while higher ranks introduce inefficiency. CoLoRA attempts to address this by introducing multiple competing LoRA components and a dynamic selection mechanism to evaluate them during training, aiming to improve model adaptability without increasing inference complexity. The primary contributions claimed by the authors include establishing a competitive learning framework for LoRA, implementing a selector to optimize component selection, and demonstrating CoLoRA’s purported advantage over conventional LoRA in tasks without additional inference overhead. However, the practical significance of these contributions remains unclear, and it is debatable whether the competitive framework provides a tangible benefit over simpler LoRA configurations. Furthermore, the experiments, while suggesting performance gains, may lack sufficient depth in real-world application scenarios to substantiate CoLoRA’s effectiveness fully.

**Strengths:**

1. Dynamic Selector Mechanism: The dynamic selection mechanism is designed to assess the performance of each LoRA component, selecting the best-performing component dynamically. This theoretically enables a more targeted model adaptation, which may contribute to achieving a balance between efficiency and expressiveness.
2. Extensive Experiments Across Benchmarks: The paper provides a variety of experimental results across benchmarks like commonsense reasoning and the MMLU, which serve to showcase the claimed advantages of CoLoRA. These evaluations attempt to establish CoLoRA’s capacity to perform well across a range of tasks, adding empirical evidence to support its potential robustness.
3. Comparison with Established Methods: CoLoRA is compared not only with standard LoRA but also with mixtures of experts (MoE) methods. This situates CoLoRA within the broader context of parameter-efficient fine-tuning techniques, allowing readers to gauge its potential advantages and limitations relative to other established approaches.

**Weaknesses:**

1. Limited novelty in competitive learning mechanism: The competitive learning framework introduced here lacks substantial novelty, as similar multi-component strategies have been explored in recent works on parameter-efficient fine-tuning. For example, recent studies on multi-component or competitive mechanisms that optimize LoRA components across tasks potentially reduce the novelty of CoLoRA’s contribution [1, 2].
2. Insufficient evaluation on practical multi-task scenarios: While the paper presents evaluations on commonsense reasoning and language understanding tasks, it lacks assessment in realistic, multi-task settings. The effectiveness of CoLoRA’s competitive learning could be better demonstrated through domain-specific applications or real-world multi-task benchmarks, as seen in recent research [3]. Adding experiments in diverse, practical applications would strengthen the paper’s claim of broad applicability.
3. Overlooked baseline comparisons: Although the paper compares CoLoRA against standard LoRA and certain MoE models, it does not include several relevant baselines that implement advanced multi-component LoRA optimizations, as presented in recent studies [2, 3]. Including these baselines would provide a more balanced evaluation, helping to demonstrate if CoLoRA truly offers unique advantages or whether existing methods meet or exceed its performance.
4. Lack of detailed analysis on selector mechanism: The dynamic selector is a core component of CoLoRA, yet there is minimal analysis on its decision-making process or robustness across task complexities. A deeper exploration of how the selector operates in various scenarios, along with sensitivity testing, would add value and practical insight to the claims of its efficacy. Such analysis could also improve the model’s interpretability and assist in refining the selector’s parameters for better performance across diverse tasks.

Ref:
[1] Q. Liu, X. Wu, X. Zhao, Y. Zhu, D. Xu, F. Tian, and Y. Zheng, "When MOE Meets LLMs: Parameter Efficient Fine-tuning for Multi-task Medical Applications."
[2] Y. Wang, Y. Lin, X. Zeng, and G. Zhang, "MultiLoRA: Democratizing LoRA for Better Multi-Task Learning."
[3] Z. Ye, D. Li, J. Tian, T. Lan, Y. Liang, Y. Jiang, J. Zuo, H. Lu, L. Duan, and M. Tang, "m-LoRA: Efficient LLM Model Fine-tune and Inference via Multi-LoRA Optimization."

**Questions:**

1. Could the authors clarify how CoLoRA differentiates itself from other competitive or multi-component frameworks, such as those seen in recent work on MultiLoRA or m-LoRA? Specifically, how does CoLoRA’s competitive learning provide unique benefits over these methods, both in terms of architecture and practical outcomes?
2. Could the authors provide more insights into how the selector’s decision-making is validated, especially under varying task complexities or noisy data environments? Additional results or qualitative analysis here could better illustrate the selector’s effectiveness and robustness.
3. How would CoLoRA perform in more practical, multi-task scenarios? Testing in a realistic multi-task setting or domain-specific application (e.g., medical or legal language tasks) would better substantiate CoLoRA’s practical utility.
4. Could the authors include or discuss comparisons with baselines like MultiLoRA and m-LoRA to ensure that CoLoRA’s claimed improvements are significant and unique?
5. How do individual LoRA components contribute to overall performance in the CoLoRA framework? A more detailed ablation or component-level analysis could reveal how each competing LoRA component affects the model’s learning process and generalization.

---

> ### Author Response · Authors · 2024-11-22
> **Official Response by Authors, Part 1**
>
> Thank you for your constructive comments. Below we have made responses to your comments. If you have any further comment, please feel free to let us know and we are more than glad to discuss with you.
>
> > Q1: Could the authors clarify how CoLoRA differentiates itself from other competitive or multi-component frameworks, such as those seen in recent work on MultiLoRA or m-LoRA? Specifically, how does CoLoRA’s competitive learning provide unique benefits over these methods, both in terms of architecture and practical outcomes?
> >
> > W1: Limited novelty in competitive learning mechanism: The competitive learning framework introduced here lacks substantial novelty, as similar multi-component strategies have been explored in recent works on parameter-efficient fine-tuning. For example, recent studies on multi-component or competitive mechanisms that optimize LoRA components across tasks potentially reduce the novelty of CoLoRA’s contribution [1, 2].
> > Ref: [1] Q. Liu, X. Wu, X. Zhao, Y. Zhu, D. Xu, F. Tian, and Y. Zheng, "When MOE Meets LLMs: Parameter Efficient Fine-tuning for Multi-task Medical Applications."
> > [2] Y. Wang, Y. Lin, X. Zeng, and G. Zhang, "MultiLoRA: Democratizing LoRA for Better Multi-Task Learning."
> > [3] Z. Ye, D. Li, J. Tian, T. Lan, Y. Liang, Y. Jiang, J. Zuo, H. Lu, L. Duan, and M. Tang, "m-LoRA: Efficient LLM Model Fine-tune and Inference via Multi-LoRA Optimization."

---

> ### Author Response · Authors · 2024-11-22
> **Official Response by Authors, Part 2**
>
> ### Answer to Q1 and W1:
>
> We think that the proposed CoLoRA is different from MoE of LoRA and the **general differences** between them can be reflected in the following aspects:
>
> 1. **Different aims**: **CoLoRA is for the training of a single LoRA** and it is primarily for single-task learning in our current work, while **MoE of LoRA [1] is to improve the performance by using multiple LoRAs**.
> 2. **Different architectures**: The main architectural difference between CoLoRA and MoE of LoRA lies in the selector of CoLoRA and the gating layer in MoE. Specifically, the **selector** in CoLoRA treats each LoRA as an individual component, **estimating its performance and providing feedback to guide the LoRA updates during the training process**. In contrast, the gating network in **MoE** **allocates the top-K experts at each layer, and forces those experts to work collectively**, with the allocation varying for each input token across different layers.
> 3. **Different training objectives**: **MoE of LoRA** optimizes the model using a single **cross-entropy loss**, while **CoLoRA** incorporates a more interactive objective. Specifically, CoLoRA applies **preferences from the language model to the selector with pairwise loss and feedback from the selector to the language model through the alignment loss**, as described in Section 3.4.
> 4. **Different inference processes**: During inference, **MoE of LoRA** selects top-K experts dynamically at each layer for every token, with routing decisions varying for each token during decoding, representing a **token-level routing mechanism**. Differently, **CoLoRA** selects the best-performing LoRA based on the selector's preference on the validation set. The selected LoRA is then merges into LLM, while **the selector and unselected LoRA are discarded**. This approach reflects a **domain-level preference**. Moreover, the inference efficiency of CoLoRA is the same as that of LoRA and it is more efficient than MoE of LoRA.
>
>
> The **detailed differences** with MoELoRA [1], MultiLoRA [2], and mLoRA [3] are presented below.
>
> Compared to **MoELoRA** [1], which relies on an MoE architecture and **requires task IDs to route specific LoRAs** during inference, CoLoRA is a training pipeline rather than an architectural modification. In CoLoRA, all LoRAs compete during training, with the winner being merged into the model, removing the need for task-specific routing and task IDs. This distinction ensures that CoLoRA is more computationally efficient during inference, as it avoids the overhead of selecting among multiple LoRAs based on task inputs.
>
> Unlike **MultiLoRA** [2], which **sums the outputs of multiple parallel LoRAs within a single layer** and optimizes them as a whole, CoLoRA incorporates a selector to dynamically estimate each LoRA's preference during training, adjusting the weight of each LoRA’s update based on its performance. In CoLoRA, the LoRAs compete against each other during training, with only the best-performing LoRA ultimately retained and used, whereas all the LoRAs in MultiLoRA are summed together after training, contributing collectively to the final model. Fundamentally, CoLoRA serves as a training pipeline, while MultiLoRA acts as a structural variant of LoRA.
>
> Regarding **mLoRA** [3], the main idea of this code repository is to **concurrently train multiple LoRAs** on a base model for different tasks, **each LoRA per task**. Its strengths lie in its **efficient implementation** and highly scalable design, allowing it to manage multiple LoRAs effectively. Essentially, **mLoRA operates like training single LoRAs but with a more efficient system** for handling multiple LoRAs. In contrast, CoLoRA focuses on competitive training, where multiple LoRAs are trained simultaneously on a single dataset and they compete based on the performance. Ultimately, in CoLoRA, only the best-performing LoRA is selected and merged into the model, and the others are discarded, eliminating the inference overhead. Unlike mLoRA that retains all LoRAs for use in different tasks, CoLoRA emphasizes the inference efficiency by using only the best one.

---

> ### Author Response · Authors · 2024-11-22
> **Official Response by Authors, Part 3**
>
> > Q2: Could the authors provide more insights into how the selector’s decision-making is validated, especially under varying task complexities or noisy data environments? Additional results or qualitative analysis here could better illustrate the selector’s effectiveness and robustness.
> >
> > W2: Insufficient evaluation on practical multi-task scenarios: While the paper presents evaluations on commonsense reasoning and language understanding tasks, it lacks assessment in realistic, multi-task settings. The effectiveness of CoLoRA’s competitive learning could be better demonstrated through domain-specific applications or real-world multi-task benchmarks, as seen in recent research [3]. Adding experiments in diverse, practical applications would strengthen the paper’s claim of broad applicability.
> ### Answer to Q2 and W2:
> Unlike the gating layer in most MoE models, **CoLoRA’s selector is not designed to route between experts during inference. Instead, it is learned during training to reflect preferences based on the LLM’s feedback (e.g., loss or performance metrics).** This enables the selector to dynamically adjust the updates for each LoRA, ensuring that the best-performing LoRA is chosen at each step. **Importantly, once training is complete, the selector is discarded, incurring no inference overhead.**
>
> To validate the selector’s decision-making, we conducted an ablation experiment by removing it during training, as described in Section 5.2. Without the selector, all LoRAs are treated equally, with no feedback-driven adjustments to their updates. As shown in **Table 5**, this leads to a significant performance drop (from 83.56 to 75.44), demonstrating that the selector is critical for enabling the competitive learning process and improving model performance. **Without the selector, there is no mechanism for adaptive competition among the LoRAs, which undermines the fine-tuning effectiveness.**
>
> Additionally, to assess the selector's robustness in noisy environments, we **injected noise with varying intensities (0–10.0)** into the pairwise loss, as described in **Section 5.6** and shown in **Table 9**. These experiments demonstrate that **CoLoRA can tolerate moderate levels of noise during training without significant performance degradation.** However, excessive noise (e.g., intensity of 10.0) adversely affects model training, indicating the limits of the selector’s resilience in highly noisy conditions.
>
> While CoLoRA is not designed as an MoE model or for multi-task learning, those ablation and noise tolerance studies provide a strong evidence of the selector's effectiveness and robustness within its intended scope. Future work could explore to extend CoLoRA to more diverse task settings or further analyze its behavior under extreme conditions.
>
> ---
> > Q3: How would CoLoRA perform in more practical, multi-task scenarios? Testing in a realistic multi-task setting or domain-specific application (e.g., medical or legal language tasks) would better substantiate CoLoRA’s practical utility.
>
> ### Answer to Q3:
>
> As mentioned in the response to Q1, CoLoRA is designed as a training pipeline that focuses on optimizing a single LoRA component through competitive learning, rather than managing multiple LoRAs across tasks. Moreover, the commonsense reasoning tasks presented in **Table 2** could serve as an example of a multi-task learning setting, **where the model is trained on a combined dataset and evaluated across eight sub-tasks without further fine-tuning. These results demonstrate CoLoRA’s strong generalization ability across diverse tasks without requiring explicit task IDs during training.**
>
> For domain-specific applications like multi-task medical tasks, as discussed in the MoELoRA paper, CoLoRA does not process task IDs or explicitly route tasks during training or inference. Instead, its selector is designed to reflect the language model’s preferences during training, making it unsuitable for task-specific adaptation. Extending CoLoRA to such domain-specific multi-task scenarios would require additional adaptations, which are beyond the scope of this work but represent a promising avenue for future exploration.

---

> ### Author Response · Authors · 2024-11-22
> **Official Response by Authors, Part 4**
>
> > Q4: Could the authors include or discuss comparisons with baselines like MultiLoRA and m-LoRA to ensure that CoLoRA’s claimed improvements are significant and unique?
> >
> > W3: Overlooked baseline comparisons: Although the paper compares CoLoRA against standard LoRA and certain MoE models, it does not include several relevant baselines that implement advanced multi-component LoRA optimizations, as presented in recent studies [2, 3]. Including baselines would provide a more balanced evaluation, helping to demonstrate if CoLoRA truly offers unique advantages or whether existing methods meet or exceed its performance.
> ### Answer to Q4 and W3:
> Thank you for the suggestion. **MoELoRA requires task IDs for training, which is not applicable in our setting, as we do not explicitly utilize task IDs.** This makes a direct comparison with MoELoRA unsuitable. Regarding **mLoRA, it operates similarly to training a single LoRA**, so its performance can be inferred from our results for LoRA in Table 2.
>
> For **MultiLoRA**, since there is no official implementation available, we **implemented our own version** based on the paper description. We trained MultiLoRA on Llama-3-8B model with learning rates of {1e-3, 1e-4, 5e-5}, selecting the best model based on the validation dataset. The performance results are as follows, where $\\#LoRA$ denotes the number of LoRAs used.
>
> | Method                           | BoolQ  | PIQA   | SIQA   | ARC-c  | ARC-e  | OBQA   | HellaS | WinoG  | Average |
> |----------------------------------|--------|--------|--------|--------|--------|--------|--------|--------|---------|
> | Multi-LoRA$_{r=4;\\#LoRA=4}$  |  61.83 |  47.33 |  32.75 |  24.15 |  26.30 |  29.20 |  25.86 |  49.49 |   37.12 |
> | Multi-LoRA$_{r=8;\\#LoRA=2}$  |  62.17 |  50.82 |  33.27 |  25.85 |  25.72 |  26.40 |  24.92 |  50.28 |   37.43 |
> | Multi-LoRA$_{r=8;\\#LoRA=4}$  |  62.17 |  50.54 |  35.16 |  23.63 |  24.75 |  23.40 |  25.05 |  50.43 |   36.89 |
> | Multi-LoRA$_{r=32;\\#LoRA=4}$ |  62.17 |  50.49 |  36.03 |  26.11 |  24.96 |  26.00 |  24.29 |  50.43 |   37.56 |
> | LoRA$_{r=4}$                     | 65.72  | 73.67  | 73.49  | 63.31  | 78.70  | 72.80  | 80.06  | 75.93  |  72.96  |
> | LoRA$_{r=8}$                     | 52.02  | 70.40  | 64.94  | 59.04  | 76.39  | 64.20  | 74.72  | 72.38  |  66.76  |
> | LoRA$_{r=16}$                    | 65.66  | 77.64  | 71.65  | 71.93  | 87.42  | 73.60  | 82.06  | 73.40  |  75.42  |
> | LoRA$_{r=32}$                    | 70.80  | 85.20  | 79.90  | 71.20  | 84.20  | 79.00  | 91.70  | 84.30  |  80.79  |
> | LoRA$_{r=128}$                   | 71.99  | 88.19  | 79.63  | 80.03  | 91.37  | 86.40  | 93.42  | 88.00  |  84.88  |
> | CoLoRA$_{K=4, r=4}$              | 69.02  | 86.56  | 79.68  | 78.84  | 92.21  | 83.40  | 92.39  | 86.35  |  83.56  |
> | CoLoRA$_{K=2, r=8}$              | 75.14  | 90.04  | 82.09  | 82.42  | 93.14  | 89.00  | 96.03  | 88.87  |  87.09  |
> | CoLoRA$_{K=4, r=8}$              | 73.79  | 88.79  | 81.47  | 83.11  | 92.68  | 89.00  | 95.45  | 88.24  |  86.57  |
> | CoLoRA$_{K=4, r=32}$             | 71.53  | 87.65  | 79.99  | 80.12  | 92.42  | 87.40  | 94.56  | 86.50  |  85.02  |
>
> As shown in the table above, MultiLoRA does not perform well on the commonsense reasoning task. This could be due to its design, which is primarily tailored for large multi-task training corpus. Additionally, the initialization method for both matrices $A$ and $B$ with a learnable scaling factor may have influenced its performance on the task. In the original MultiLoRA paper, the authors trained their models on significantly larger datasets, including Alpaca, MMLU, GSM8K, and SuperGLUE. In contrast, the commonsense reasoning dataset used here contains only 170k entries, which may have led MultiLoRA to overfit to the smaller dataset.
>
> Unlike MultiLoRA, CoLoRA is not explicitly designed for multi-task learning scenarios. Instead, it focuses on delivering more generalized performance, making it better suited for diverse task settings.

---

> ### Author Response · Authors · 2024-11-22
> **Official Response by Authors, Part 5**
>
> > Q5: How do individual LoRA components contribute to overall performance in the CoLoRA framework? A more detailed ablation or component-level analysis could reveal how each competing LoRA component affects the model’s learning process and generalization.
> >
> > W4: Lack of detailed analysis on selector mechanism: The dynamic selector is a core component of CoLoRA, yet there is minimal analysis on its decision-making process or robustness across task complexities. A deeper exploration of how the selector operates in various scenarios, along with sensitivity testing, would add value and practical insight to the claims of its efficacy. Such analysis could also improve the model’s interpretability and assist in refining the selector’s parameters for better performance across diverse tasks.
>
> ### Answer to Q5 and W4:
> In the CoLoRA framework, individual LoRA components play a crucial role in the overall performance by competing with each other during training, which ultimately leads to better performance compared to traditional single-LoRA training approaches. As detailed in **Section 5.1**, **Figure 2**, and **Figure 3**, this competitive learning process enables the LoRA components to **collaborate in minimizing the overall loss**, driving the model toward a more efficient and robust solution. Each LoRA component is trained independently and then selected based on its performance, ensuring that the most effective LoRA is retained.
>
> Additionally, we provided a more **detailed analysis in Section 5.7 and Table 10**, where we examine **the performance of each LoRA component** after the CoLoRA training process. The results clearly demonstrate that **all LoRA components achieve improvements** in performance compared to conventional LoRA training, highlighting the effectiveness of the competitive training pipeline. This indicates that the collaborative competition between LoRAs results in strong LoRAs.

---

> > ### Comment · Reviewer_RpHx · 2024-11-25
> >
> > Thank you for your rebuttal. I have decided to increase my score to 5.

---

> > > ### Author Response · Authors · 2024-11-25
> > >
> > > Thank you for taking the time to review our rebuttal and for reconsidering your assessment of our work. We greatly appreciate your thoughtful feedback and the opportunity to address your concerns. If you have any additional comments or suggestions, we would be happy to address them further.

---

### Meta-Review · Area_Chair_Fjcw · 2024-12-23

**Metareview:**

The paper proposes CoLoRA, an extension of the existing LoRA approach. Key innovation includes using multiple models instead of a single LoRA.

The authors present detailed experiments both in the main text and rebuttal and highlight the advantages. The biggest criticisms are from one reviewer on presentation, computational complexity among other things.

My recommendation as a poster is based on a general positive outlook from most of the reviewers and the incremental nature of the contribution.

**Additional Comments On Reviewer Discussion:**

The authors provide a very detailed rebuttal to the reviews and manage to convince some of the reviewers to increase the score. From the discussion (or rather lack of), I have the sense that while the overall sentiment is positive, none of the reviewers feel strongly about the paper.

---

### Decision · Program_Chairs · 2025-01-22

Accept (Poster)